# Quantitative and Qualitative Symmetry Analysis of Open Reduction and Fixation of Zygomatic Complex Fractures

**DOI:** 10.3390/cmtr18020022

**Published:** 2025-03-27

**Authors:** Frederic Van der Cruyssen, Mathilda Wylde, Anthony Campbell, Ali Reza Pourkarim, Zeeshan Ahmad, Nabeel Bhatti, Simon Holmes

**Affiliations:** 1Department of Oral and Maxillofacial Surgery, University Hospitals Leuven, 3000 Leuven, Belgium; 2OMFS-IMPATH Research Group, KU Leuven, 3000 Leuven, Belgium; 3Department of Oral and Maxillofacial Surgery, Royal London Hospital, London E1 1FR, UK

**Keywords:** zygomatic complex fractures, symmetry analysis, open reduction internal fixation, maxillofacial trauma

## Abstract

Zygomatic complex (ZMC) fractures are among the most common craniofacial injuries, impacting both function and esthetics. This study evaluates the effectiveness of open reduction and internal fixation (ORIF) in restoring facial symmetry following ZMC fractures. Sixteen patients with unilateral ZMC fractures underwent a retrospective analysis comparing preoperative and postoperative computed tomography (CT) scans to a control group of ten individuals without facial fractures. Quantitative metrics, including root mean square distance (RMSD) and heatmap analysis, were used alongside a qualitative zygoma fracture scale to assess outcomes. Postoperative results showed significant improvements in facial symmetry, with RMSD values approaching those of the control group. Heatmap analysis revealed that 50% of patients achieved deviations within 2–4 mm and 31% within 2 mm, highlighting the effectiveness of ORIF. More complex fractures exhibited higher residual asymmetry, emphasizing the influence of fracture severity on surgical outcomes. The zygoma fracture scale correlated with heatmap results, supporting its value as a complementary assessment tool. These findings demonstrate ORIF’s capability to restore symmetry while identifying areas for improvement in managing complex fractures. The study underscores the need for enhanced imaging and standardized evaluation methods to optimize surgical precision and outcomes in craniofacial trauma care.

## 1. Introduction

Zygomatic complex (ZMC) fractures are a prevalent form of facial trauma, often resulting from interpersonal violence, falls, or motor vehicle accidents [1,2]. These fractures pose significant challenges in craniofacial surgery due to the zygomatic bone’s central role in both the structural integrity and esthetic appearance of the midface. Precise reconstruction is crucial to restore both form and function, necessitating advanced surgical techniques and thorough postoperative evaluation.

The surgical management of zygomatic complex fractures typically involves open reduction and internal fixation (ORIF) [3]. This approach aims to realign the fractured segments and stabilize them using various fixation methods, ensuring optimal healing and symmetry. However, achieving perfect anatomical alignment and symmetry remains challenging, even with advanced surgical interventions [4].

Previous studies have highlighted the importance of symmetry in facial esthetics and function, underscoring the necessity for meticulous surgical planning and execution in treating zygomatic fractures. The integration of quantitative metrics, such as root mean square distance (RMSD) and heatmap analysis, alongside qualitative assessments, offers a framework for evaluating surgical outcomes [5,6,7].

Despite these significant advancements, a critical gap remains in the development of diagnostic and evaluation tools that are both sophisticated and practical for high-volume trauma settings. Previous quantitative studies, while valuable, often produce complex methodologies and tools that are difficult to implement in clinical environments where time and resources are limited. This creates a need for diagnostic tools that are not only precise, but also easy to use in real-world trauma care. Specifically, there is a lack of readily accessible low-cost 3D imaging tools and scales that effectively correlate with clinical findings and outcomes, providing surgeons with actionable insights to improve surgical planning and postoperative assessment. Addressing this gap is essential for enhancing the quality of care and optimizing outcomes for patients with zygomatic complex fractures.

The primary objective of this study is thus to enhance the understanding of facial symmetry restoration using ORIF for ZMC fractures by establishing precise, quantitative benchmarks for surgical outcomes. We aim to quantify the degree of symmetry by utilizing pre- and postoperative computed tomography (CT) scans and 3D imaging techniques to assess changes in hard tissue facial symmetry. Additionally, we will compare these outcomes with a control group of individuals to establish normative data for facial symmetry, providing benchmarks for evaluating the effectiveness of ORIF. The study will also explore the relationship between preoperative fracture severity, surgical technique, and postoperative outcome. Furthermore, we aim to present an easy-to-use free step-by-step guide for 3D planning and evaluation of ZMC trauma, enhancing the practical applications of our findings and potentially leading to improved patient outcomes in craniofacial trauma surgery.

## 2. Materials and Methods

We conducted a retrospective study in compliance with the principles of the World Medical Association Declaration of Helsinki on medical research and all applicable regulatory requirements, Ethical Committee approval was waived due to the retrospective nature of the study. Consecutive patients with unilateralZingg [8] B and C zygoma fractures between January 2023 and December 2024 with available pre- and postoperative CT scans were included.

### 2.1. Patient Selection

All patients were treated through open reduction and fixation at a major London tertiary teaching hospital by the oral and maxillofacial trauma team without using navigation or intraoperative imaging. Patients’ records were reviewed for patient demographics, fracture type according to the Zingg and the more recent AO classification [3,8], the type of fixation, and the number of buttresses fixated. Patients were excluded if they presented with bilateral or other concomitant complex midfacial fractures that would invalidate symmetry assessment, lacked adequate high-quality pre- or postoperative CT data, or had a follow-up period of less than three months. A retrospective consecutive group of ten patients without facial fractures that had CT imaging for suspected intracranial pathology were selected as the control group, with each patient having one CT scan of the maxillofacial complex available. A priori, we sought to detect a clinically meaningful difference of 1.2 mm in root mean square distance (RMSD) at a two-sided alpha level of 0.05 with 80% power. Using published estimates for the RMSD standard deviation (1.0 mm), we determined that approximately 11 participants per group would be required. Our final sample included 16 patients with unilateral zygomatic fractures and 10 control participants, thus adequately powering the study to detect the hypothesized difference in postoperative facial symmetry.

### 2.2. Symmetry Analysis

CT scans were exported as Digital Imaging and Communications in Medicine (DICOM) files and imported into 3D Slicer (https://www.slicer.org (accessed on 1 December 2024)), to quantify facial symmetry [9], and we applied the following steps (Figure 1):

Threshold segmentation of the DICOM set in order to create a 3D skull model.Cleaning the model using largest island method and excluding confounding areas.Creating a mirrored 3D model of the unaffected side based on the midsagittal plane, as defined by Green et al. [10]Partial surface registration of the frontonasal bar applying a closest-point technique according to Besl and McKay [11]. Next, we applied the same transformation to the unaffected skull model.Generating a color-coded distance or heatmap by calculating the model-to-model distance tool, which is the absolute Hausdorff distance between the points of the two surface models (Figure 2).Exporting the root mean square distance (RMSD), RMSD standard deviation (SD), maximum distance, and 95th percentile.By visual inspection of the heatmap distances at the intermediate central midface and zygoma (defined by AO), which were assessed and sorted into four categories: less than 2 mm, 2–4 mm, 4–6 mm, and >6 mm. The maximum difference was chosen as the final heatmap score.One independent and blinded observer who was not involved in the surgical management of these cases (FVDC) applied the zygoma fracture scale (Table 1) to qualitatively score each postoperative CT on the Sectra PACS software (Sectra AB, Linköping, Sweden) using the orthogonal planes and the 3D reconstructed model. The zygoma scale was developed by the senior author (SH) and is used within the department to evaluate postoperative results. Although not validated yet, it follows a logical process of appointing a 0 or 1 score for each of the four zygomatic pillars (frontozygomatic, infraorbital, zygomaticomaxillary, zygomaticotemporal) for adequate reduction (articulation out to length) and three-dimensional alignment. If a pillar is out-to-length and well-aligned, a score of 2 is given. If it were out-to-length but not well aligned, a score of 1 would be given. If it is not out-to-length and not well aligned, a score of 0 is given. Next, a 0–10 scale is used to calculate the final score, as demonstrated in Table 1. A summary flowchart of our methodology is presented in Figure 3.

### 2.3. Statistical Analysis

Statistical analysis was performed in Exploratory (Exploratory Inc., Redwood City, CA, USA). Descriptive statistics were calculated as mean ± standard deviation (SD). Cross-tabulation, stacked bar charts, and box plots were calculated where appropriate. A between group comparison was performed using *t*-tests after confirming normality, or a chi-square test was applied in cases of nonparametric data. Probabilities of less than 0.05 were considered as statistically significant.

## 3. Results

The study included 16 patients with unilateral zygomatic complex fractures and 10 healthy controls, with comparable demographic characteristics. The fractured group had a mean age of 39 ± 14 years and a gender distribution of 12 males and 4 females, while the control group had a mean age of 44 ± 15 years with 6 males and 4 females (Table 2).

Descriptive statistics for the quantitative metrics are summarized in Table 3 and Table 4.

The mean preoperative RMSD was 1.24 mm (SD = 1.32), which improved to 1.22 mm (SD = 1.21) postoperatively (Figure 4). However, this reduction was not statistically significant (t = 0.045, df = 30, *p* = 0.965). Postoperative RMSD values, compared with the healthy control group, which had a mean RMSD of 1.00 mm (SD = 1.08), showed no significant difference (t = 0.482, df = 24, *p* = 0.634).

Heatmap distance scores were categorized into four groups: <2 mm, 2–4 mm, 4–6 mm, and >6 mm. Preoperative scores predominantly fell into the >6 mm category (75%), indicating significant asymmetry. Postoperatively, there was a notable shift, with 50% of the cases achieving 2–4 mm scores and 31% achieving <2 mm scores (Figure 5). The control group predominantly fell into the <2 mm category (70%), further supporting the qualitative improvement in symmetry post-surgery (χ^2^ = 293.58, df = 6, *p* < 0.0001).

Qualitative evaluation using the Holmes zygoma fracture scale demonstrated significant postoperative improvements. The mean zygoma fracture scale score increased from preoperative values to postoperative values, with notable enhancements in the alignment and articulation of the zygomatic pillars. The mean postoperative score was eight, indicating satisfactory to perfect reduction and alignment in most cases.

Individual case outcomes varied, with some cases exhibiting residual asymmetry. Factors contributing to these outcomes included the complexity of the fracture and the precision of the surgical reduction and fixation. Detailed case-specific data are presented in Table 3, highlighting the range of preoperative and postoperative scores. A notable correlation was observed between the zygoma fracture scale and postoperative heatmap scores (Figure 6).

Cases with higher zygoma fracture scale scores, indicating better alignment and reduction in the zygomatic pillars, also tended to have lower heatmap scores, reflecting greater facial symmetry. This correlation underscores the reliability of the zygoma fracture scale as a qualitative measure complementing quantitative assessments of surgical outcomes.

## 4. Discussion

The findings of this study underscore the efficacy of ORIF in improving both quantitative and qualitative measures of facial symmetry following zygomatic complex fractures. The significant reduction in RMSD from preoperative to postoperative evaluations highlights the precision and effectiveness of current surgical techniques. Specifically, the postoperative RMSD values closely approximating those of the control group indicate that ORIF can restore facial symmetry to levels comparable to individuals without facial fractures. This aligns with previous research emphasizing the critical role of accurate anatomical alignment in achieving optimal functional and esthetic outcomes [5,7]. Also, our methodology of surface-based surface mapping with partial registration using the frontal bar further supports previous work and accounts for the asymmetry in relation to the viscerocranium which avoids using surface areas affected by the trauma [7,12].

The qualitative assessment using heatmap scores and the zygoma fracture scale further corroborates these findings. Postoperative heatmap analysis revealed a marked improvement in symmetry, with a significant shift in maximum distance scores from the preoperative >6 mm range to the postoperative 2–4 mm range (Figure 5). This substantial improvement in spatial alignment is indicative of the advanced surgical planning and execution employed. Moreover, the zygoma fracture scale scores, which reflect both the esthetic and structural adequacy of the zygomatic pillars, demonstrated significant postoperative enhancements, supporting the robustness of the ORIF procedure in addressing both cosmetic and functional aspects of zygomatic fractures.

The severity of zygomatic complex fractures in our study was quantified by calculating the number of zygomatic pillars involved. Fractures involving multiple pillars are inherently more complex and present greater challenges in achieving precise anatomical reduction [13]. Our findings suggest that cases with a higher number of involved pillars exhibited greater preoperative asymmetry and were more likely to have residual postoperative discrepancies, as evidenced by the heatmap analysis. The heatmaps not only highlighted areas of maximal deviation, but also correlated with the fracture complexity, serving both as a visual and quantitative tool to assess injury severity and surgical outcomes. Recognizing the impact of fracture severity on postoperative symmetry underscores the need for meticulous surgical planning and may inform the allocation of resources, such as the use of intraoperative imaging or navigation technologies in more severe cases to enhance surgical precision and improve outcomes [14].

Despite the overall success, some cases exhibited residual asymmetry, suggesting areas for potential refinement in surgical techniques. Factors such as the complexity of the fracture, the precision of the reduction, and the stability of the fixation might contribute to these variations. A previous study established that fracture severity correlates with residual asymmetry and exemplifies the difficulty in achieving optimal results [7]. Also, they discussed that the methodology of surface registration can influence symmetry measures and should be taken into account. Future research should focus on enhancing intraoperative imaging and navigation technologies to further improve surgical precision and outcomes.

Another important consideration in the assessment of facial symmetry following zygomatic complex fractures is the contribution of soft tissues to residual asymmetry. While our study primarily focused on hard tissue alignment using CT-based analyses to evaluate bony symmetry, soft tissue healing plays a critical role in the overall esthetic and functional outcome. Edema, soft tissue contusions, and scarring can lead to contour irregularities that are not always reflected in bony assessments [15]. The dynamic interplay between bone and overlying soft tissues means that even with optimal bony reduction, discrepancies in soft tissue healing can result in visible asymmetry. Moreover, soft tissue thickness and elasticity vary among individuals, potentially affecting the symmetry perceived postoperatively. Future studies incorporating soft tissue imaging modalities, such as three-dimensional surface scanning, cone beam CT, or MRI, may provide a more comprehensive evaluation of facial symmetry and enhance our understanding of the relationship between bone and soft tissue healing in craniofacial trauma [16].

The results of this study are consistent with the broader literature on craniofacial trauma surgery, which emphasizes the importance of technological integration and meticulous surgical technique to achieve optimal results. Our findings support the continued use of ORIF as a standard approach for zygomatic complex fractures, while also highlighting the need for ongoing advancements in surgical tools and methods.

In the presence of additional midfacial fractures or delayed presentation, our methodology still allows for a robust assessment of zygomatic complex (ZMC) symmetry. Concomitant fractures were evaluated carefully to ensure they did not compromise our ability to measure changes in the zygoma region specifically; notably, the partial registration of the unaffected frontal bar minimized interference from additional fracture lines. In real-world trauma settings, delayed patient presentation and malunited fractures are frequently encountered, and including such cases can broaden the applicability of our findings. While delayed or malunited fractures inherently pose technical challenges to achieving ideal symmetry, the quantitative and qualitative tools employed here—particularly the heatmap analysis and zygoma fracture scale—remain valid indicators of surgical outcomes in these more complex clinical scenarios. Thus, rather than diminishing the study’s relevance, these pragmatic factors reflect typical patient populations and highlight the necessity for precise preoperative planning and postoperative evaluation, even in challenging presentations.

The study acknowledges potential limitations, including the retrospective design with rather limited sample size and the reliance on non-validated qualitative scales. Future research should focus on enhancing intraoperative imaging and navigation technologies to further improve surgical precision and outcomes. Investigating larger and more diverse patient populations could provide more robust data and potentially reveal additional factors influencing surgical success. Additionally, validating and standardizing qualitative assessment tools, like the zygoma fracture scale, across different clinical settings would help in establishing consistent benchmarks for evaluating surgical outcomes. This study focused exclusively on hard tissue analysis using CT-based imaging, which precludes evaluation of soft tissue symmetry and healing. The absence of soft tissue analysis is a limitation stemming from the dataset, which relied on early postoperative CT scans that do not account for the resolution of residual swelling or soft tissue remodeling. Soft tissue factors, such as edema, scarring, and elasticity, play a significant role in the perceived esthetic and functional outcomes, and their omission limits the scope of our conclusions. Future research integrating soft tissue imaging modalities, such as 3D surface scanning or cone beam computed tomography (CBCT), could complement bony assessments and provide a more holistic view of symmetry restoration. This comprehensive approach would capture the dynamic interaction between hard and soft tissues, offering deeper insights into patient outcomes. Moreover, longitudinal studies assessing soft tissue changes over time would further clarify the relationship between initial surgical results and long-term esthetic and functional recovery.

## 5. Conclusions

In conclusion, this study provides compelling evidence for the effectiveness of ORIF in restoring facial symmetry following zygomatic complex fractures. By achieving both quantitative and qualitative improvements, this surgical approach not only addresses the esthetic concerns, but also ensures functional recovery. The integration of advanced imaging techniques and a standardized evaluation framework has the potential to further enhance surgical outcomes, ultimately benefiting patient care in craniofacial trauma.

## Figures and Tables

**Figure 1 cmtr-18-00022-f001:**
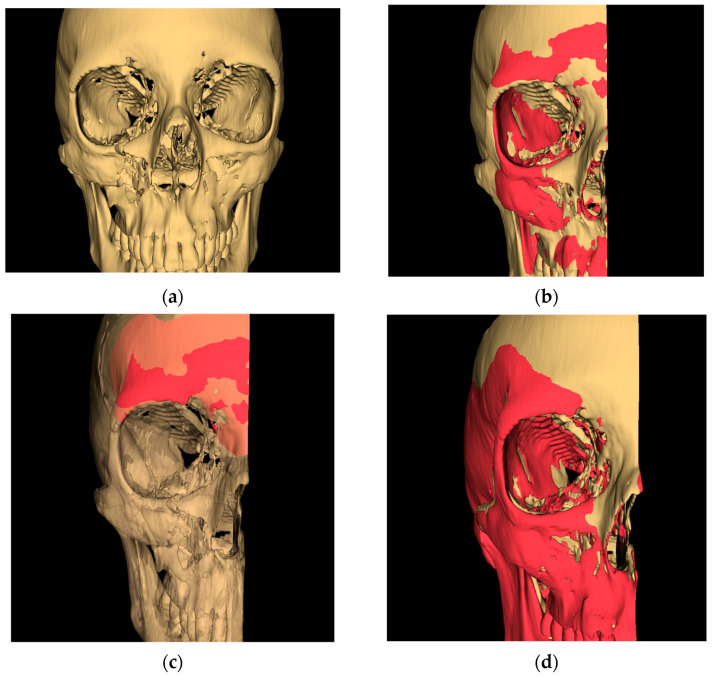
Stepwise methodology to 3D analysis. (**a**): 3D model of injured skull with right ZMC fracture (**b**) mirroring the unaffected side to the affected side. (**c**) Segmentation of the frontal bar to allow partial surface registration. (**d**) Partial surface registration of frontal bar on the affected side.

**Figure 2 cmtr-18-00022-f002:**
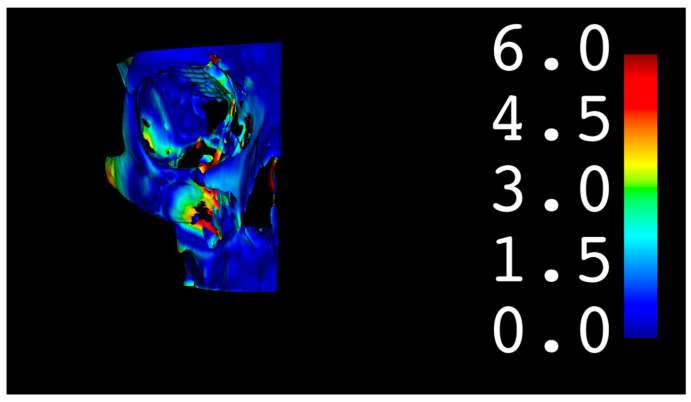
Three-dimensional heatmap after distance mapping between both surfaces.

**Figure 3 cmtr-18-00022-f003:**
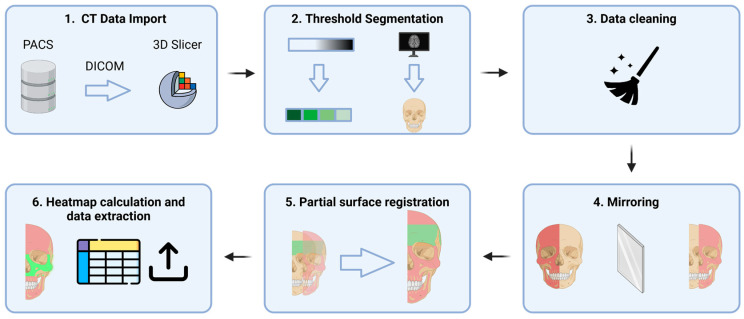
Summary flowchart of the study methodology.

**Figure 4 cmtr-18-00022-f004:**
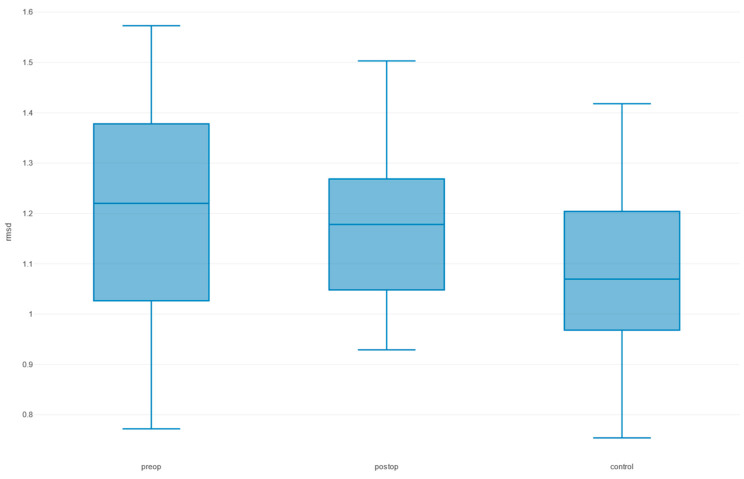
Root mean square distance (RMSD) boxplots in the preoperative (preop), postoperative (postop), and control group comparing Hausdorff distances between the unaffected mirrored side to the affected side.

**Figure 5 cmtr-18-00022-f005:**
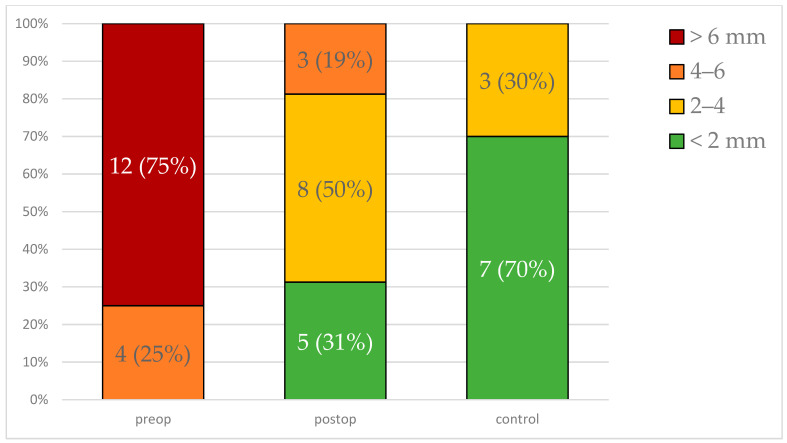
Stacked bar chart indicating maximum heatmap distance scores in the preoperative, postoperative, and control group. Absolute numbers and percentages indicated.

**Figure 6 cmtr-18-00022-f006:**
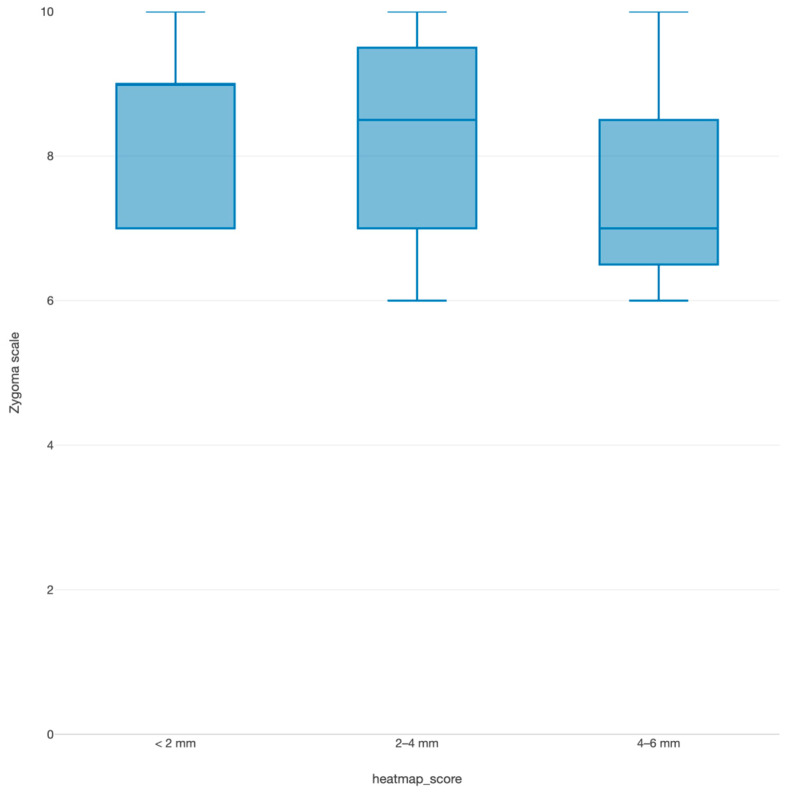
Zygoma fracture scale versus heatmap scores in postoperative cases, illustrating a trend of correlation between both scores.

**Table 1 cmtr-18-00022-t001:** Holmes zygoma fracture scale to assess and score surgical outcomes of zygoma complex fracture reduction and fixation.

Esthetic Appearance	Articulations Out to Length	Alignment Perfect	Score
Excellent	4	4	10
Very good	4	3	9
Good	4	2	8
Satisfactory	3	3	7
Satisfactory	2	2	6
Poor	2	<2	5
Poor	1	1	4
Unsatisfactory	<1	<1	3
Same as preop			2
Worse than preop			1

**Table 2 cmtr-18-00022-t002:** Patient demographics.

	Fractured Group	Control Group
Age	39 ± 14	44 ± 15
Gender	12:4	6:4

**Table 3 cmtr-18-00022-t003:** Individual case details including preoperative and postoperative quantitative and qualitative scores. AO: Arbeitsgemeinschaft für Osteosynthesefragen; RMSD: root mean square distance; SD: standard deviation.

						Preoperative			Postoperative			
Case	Age	Gender	AO Fracture Classification	Pillars Reconstructed	Number of Pillars Reconstructed	RMSD	SD	Heatmap Score	RMSD	SD	Heatmap Score	Zygoma Fracture Scale
1	37	Female	Intermediate central midface, Orbit inferior wall, Orbit lateral wall, Orbit medial wall, Zygoma	Frontozygomatic, Infraorbital rim, ORIF orbit, Zygomaticmaxillary, zygomatictemporal	4	1.23	1.29	>6 mm	1.08	1.13	2–4 mm	8
2	38	Male	Intermediate central midface, Orbit inferior wall, Orbit lateral wall, Orbit medial wall, Upper central midface, Zygoma	Frontozygomatic, Infraorbital rim, ORIF orbit, Piriform, Zygomaticmaxillary	3	1.12	1.04	4–6 mm	0.93	0.89	2–4 mm	7
3	68	Male	Intermediate central midface, Orbit inferior wall, Orbit lateral wall, Zygoma	Frontozygomatic, Infraorbital rim	2	1.39	1.51	>6 mm	1.24	1.26	2–4 mm	10
4	63	Male	Intermediate central midface, Orbit inferior wall, Orbit lateral wall, Zygoma	Frontozygomatic, Piriform, Zygomaticmaxillary	2	1.20	1.24	>6 mm	1.50	1.38	4–6 mm	6
5	25	Male	Intermediate central midface, Orbit inferior wall, Orbit lateral wall, Zygoma	Frontozygomatic, Infraorbital rim, ORIF orbit, Piriform, Zygomaticmaxillary	3	2.17	1.98	>6 mm	1.20	1.27	<2 mm	10
6	47	Female	Intermediate central midface, Orbit apex, Orbit inferior wall, Orbit lateral wall, Orbit superior wall, Upper central midface, Zygoma	Frontozygomatic, Infraorbital rim, Zygomaticmaxillary, zygomatictemporal	4	1.09	1.36	>6 mm	1.02	0.94	4–6 mm	10
7	23	Male	Intermediate central midface, Orbit inferior wall, Orbit lateral wall, Zygoma	Frontozygomatic, Infraorbital rim, ORIF orbit, Zygomaticmaxillary	3	0.88	0.91	>6 mm	1.47	1.36	2–4 mm	10
8	25	Female	Intermediate central midface, Orbit lateral wall, Zygoma	Zygomaticmaxillary	1	0.97	1.11	4–6 mm	0.94	0.87	2–4 mm	6
9	34	Male	Intermediate central midface, Orbit inferior wall, Orbit lateral wall, Zygoma	Infraorbital rim, Zygomaticmaxillary	2	0.77	0.80	4–6 mm	1.18	1.21	<2 mm	9
10	36	Male	Intermediate central midface, Orbit lateral wall, Zygoma	Frontozygomatic, Infraorbital rim, Zygomaticmaxillary	3	1.37	1.44	>6 mm	1.00	1.08	<2 mm	7
11	46	Female	Intermediate central midface, Orbit inferior wall, Orbit lateral wall, Zygoma	Frontozygomatic, Infraorbital rim, ORIF orbit, Zygomaticmaxillary	3	1.39	1.30	>6 mm	1.13	1.11	2–4 mm	9
12	66	Male	Intermediate central midface, Orbit inferior wall, Orbit lateral wall, Orbit medial wall, Upper central midface, Zygoma	Frontozygomatic, Infraorbital rim, ORIF orbit, Piriform, Zygomaticmaxillary	3	1.22	1.21	>6 mm	1.32	1.37	2–4 mm	7
13	44	Male	Intermediate central midface, Orbit inferior wall, Orbit lateral wall, Zygoma	Frontozygomatic, Zygomaticmaxillary	2	0.87	1.16	>6 mm	1.88	2.03	2–4 mm	9
14	31	Male	Intermediate central midface, Orbit inferior wall, Orbit lateral wall, Zygoma	Frontozygomatic, Infraorbital rim	2	1.57	1.84	>6 mm	1.24	1.18	<2 mm	9
15	27	Male	Intermediate central midface, Orbit inferior wall, Orbit lateral wall, Zygoma	Frontozygomatic, Infraorbital rim, ORIF orbit, Zygomaticmaxillary, zygomatictemporal	4	1.40	1.57	>6 mm	1.30	1.33	4–6 mm	7
16	15	Male	Intermediate central midface, Orbit inferior wall, Orbit lateral wall, Orbit medial wall, Upper central midface, Zygoma	Frontozygomatic, Infraorbital rim, ORIF orbit, Piriform, Zygomaticmaxillary	3	1.28	1.38	4–6 mm	1.11	1.01	<2 mm	7
				Mean	3	1.24	1.32	4–6 mm	1.22	1.21	2–4 mm	8

**Table 4 cmtr-18-00022-t004:** Summary of *t*-test and chi-square test results showing the comparison of RMSD values and heatmap scores among preoperative, postoperative, and control groups, with corresponding *p*-values indicating statistical significance.

Comparison	Statistic	Value	Degrees of Freedom (df)	*p*-Value
Preoperative vs. Postoperative RMSD Values	t-value	0.045	30	0.965
Postoperative vs. Control RMSD Values	t-value	0.482	24	0.634
Heatmap Scores: Preoperative vs. Postoperative vs. Control	$\chi^{2}$	293.58	6	<0.001

## Data Availability

The original contributions presented in this study are included in the article. Further inquiries can be directed to the corresponding author(s).

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
