# Peer review of "Quantitative and Qualitative Symmetry Analysis of Open Reduction and Fixation of Zygomatic Complex Fractures"

_1943-3883, 2025, doi:10.3390/cmtr18020022_

Round 1
Reviewer 1 Report
Comments and Suggestions for Authors
The manuscript is well written. However, a statement of presentation to the Institutional Ethical Committee or Institutional Review Board needs to be incorporated into the methodology. How was the sample size chosen? Were there any exclusion criteria? The relevance of the study in the presence of concomitant fractures, delayed presentation of the patients, and malunited fractures can be discussed.
Author Response
Reviewer 1:
The manuscript is well written. However, a statement of presentation to the Institutional Ethical Committee or Institutional Review Board needs to be incorporated into the methodology.
How was the sample size chosen? Were there any exclusion criteria?
The relevance of the study in the presence of concomitant fractures, delayed presentation of the patients, and malunited fractures can be discussed.
Response to reviewer 1:
EC approval was waived due to the retrospective nature of this study. This was added to the methods section. Our sample size calculation was added to paragraph 2.1.
Also, we elaborated on exclusion criteria in the Patient selection paragraph.
Thank you for highlighting the importance of discussing concomitant fractures, delayed presentation, and malunited fractures. We have added a dedicated paragraph in the Discussion to elaborate on how these real-world clinical factors can affect treatment planning and outcome evaluation. Specifically, we clarify that our partial surface registration technique and quantitative analysis methods remain robust even in the presence of additional midfacial injuries, and we emphasize the broader applicability of our findings in typical clinical practice where delayed or malunited fractures are often encountered.
Reviewer 2 Report
Comments and Suggestions for Authors
I suggest using the abbreviation ZC for the name zygomatic complex, because the use of ZMC suggests the term zygomaticomaxillary complex, or change terminology.
In the discussion, (row 185), you wrote about a significant reduction in RMSD, but we observed that this reduction is not statistically significant. Additionally, in 5 out of 16 patients, RMSD was higher postoperatively, which is 31%, suggesting that it may not be a good quantitative method. It is also possible that the number of samples is too small, and should be larger, but you already mentioned that in the disscussion as deficiency of the study.
Heatmap analysis seems very interested, logical and useful.
Author Response
Reviewer 2:
Interesting study that shows us some new mathematical models that could be implemented in the choice and type of treatment of complex zygomatic fractures as well as other fractures of the facial skeleton.
Response to reviewer 2:
Thank you for your positive feedback and for recognizing the potential broader applications of our proposed mathematical models. We share your enthusiasm about extending these methods to other facial fractures and believe they may offer a more standardized, quantitative framework for surgical planning and assessment.
Reviewer 3 Report
Comments and Suggestions for Authors
Very well written and outlined article. Up-to-date references. Retrospective analysis of zygomatic fracture reconstruction using segmentation and mirroring with 3d software. Study verifying the quality and quantitative of the positioning and results of the fixation buttress .
Study design and methodology very well applied. My only suggestion would be to show the flowchart for creating the composite skull and the analysis carried out at each point.
I recommend publication without revisions. This article is highly relevant fortraining centers in maxillofacial surgery.
Author Response
Reviewer 3:
Very well written and outlined article. Up-to-date references.
Retrospective analysis of zygomatic fracture reconstruction using segmentation and mirroring with 3D software.
Study verifying the quality and quantitative positioning and results of the fixation buttress.
Study design and methodology very well applied. My only suggestion would be to show the flowchart for creating the composite skull and the analysis carried out at each point.
Response to reviewer 3:
Thank you very much for your positive assessment of our article and the relevance of our references. We appreciate your suggestion to include a flowchart illustrating the composite skull creation and associated analytical steps. Following your recommendation, we have added a flowchart (Figure 3) in the revised manuscript (see “Methods” section), which clearly presents each stage—ranging from CT image segmentation and mirroring to partial surface registration and final symmetry analysis. We hope that this visualized workflow will further enhance clarity and replicate our methodology in future research.